# Evaluation of the Growth-Inhibitory Spectrum of Three Types of Cyanoacrylate Nanoparticles on Gram-Positive and Gram-Negative Bacteria

**DOI:** 10.3390/membranes12080782

**Published:** 2022-08-15

**Authors:** Fean Davisunjaya Sarian, Kazuki Ando, Shota Tsurumi, Ryohei Miyashita, Koichi Ute, Takeshi Ohama

**Affiliations:** 1School of Environmental Science and Engineering, Kochi University of Technology, 185 Miyanokuchi, Tosayamada, Kami 782-8502, Japan; 2Department of Applied Chemistry, Tokushima University, 2-1 Minami-Josanjima, Tokushima 770-8506, Japan

**Keywords:** antibacterial agent, cyanoacrylate nanoparticle, membrane damage, reactive oxygen species, stress response

## Abstract

The development of novel effective antibacterial agents is crucial due to increasing antibiotic resistance in various bacteria. Poly (alkyl cyanoacrylate) nanoparticles (PACA-NPs) are promising novel antibacterial agents as they have shown antibacterial activity against several Gram-positive and Gram-negative bacteria. However, the antibacterial mechanism remains unclear. Here, we compared the antibacterial efficacy of ethyl cyanoacrylate nanoparticles (ECA-NPs), isobutyl cyanoacrylate NPs (*^i^*BCA-NPs), and ethoxyethyl cyanoacrylate NPs (EECA-NPs) using five Gram-positive and five Gram-negative bacteria. Among these resin nanoparticles, ECA-NPs showed the highest growth inhibitory effect against all the examined bacterial species, and this effect was higher against Gram-positive bacteria than Gram-negative. While *^i^*BCA-NP could inhibit the cell growth only in two Gram-positive bacteria, i.e., *Bacillus subtilis* and *Staphylococcus aureus*, it had negligible inhibitory effect against all five Gram-negative bacteria examined. Irrespective of the differences in growth inhibition induced by these three NPs, N-acetyl-L-cysteine (NAC), a well-known reactive oxygen species (ROS) scavenger, efficiently restored growth in all the bacterial strains to that similar to untreated cells. This strongly suggests that the exposure to NPs generates ROS, which mainly induces cell growth inhibition irrespective of the difference in bacterial species and cyanoacrylate NPs used.

## 1. Introduction

Despite the abundance of antibacterial agents, bacterial infections remain a major public health problem that is exacerbated by the overuse of antibiotics. This leads to outbreaks of multi-drug resistant bacteria (MDR) [1,2], which can occur via various pathways [3,4].

In recent years, several nanoparticles (NPs) and their derivatives have gained attention because of their antibacterial activity, making them potentially useful for medical applications [5,6,7]. The antibacterial activity of NPs depends on various factors such as their chemical composition, shape, and size [8]. With significant developments in the understanding of nanosystems, some strategies to enhance NPs’ activity have been assessed, including combining treatment of NPs and irradiation [9], coating NPs with several coating agents [10], and conjugating NPs with antibiotics [11,12]. Among the widely employed nanoparticles to date, silver nanoparticles (Ag-NPs) have been significantly researched [13,14,15]. However, several in vitro studies indicated that Ag-NPs have potentially harmful effects on human health [16,17].

Poly (alkyl cyanoacrylate) nanoparticles (PACA-NPs) are used mostly to produce adhesives, surgical glue, and drug carrying nanocapsules [18,19,20]. Recently, however, cyanoacrylate nanoparticles have been increasingly studied as potential antibiotic alternatives against bacteria [21]. Moreover, PACA-NPs have been reported to inhibit the growth of a green microalga, *Chlamydomonas reinhardtii* [22] and several non-green algae [23].

Generally, targets of antibacterial drugs are intracellular such as ribosomal proteins and transcription factors. Reports show that the cell walls of *Escherichia coli* and *Bacillus subtilis* are permeable to particles below a 2 nm diameter [24]. However, most engineered resin NPs typically much larger than 2 nm and hence, cannot pass through. Therefore, these resin NPs induce their antibacterial effects using a mechanism completely different from that of antibiotics.

In this study, the antibacterial mechanism of three PACA-NPs (ethyl-, methyl-, and ethoxyethyl-cyanoacrylate polymers) at various concentrations were investigated against several Gram-positive and Gram-negative bacteria.

## 2. Materials and Methods

### 2.1. Materials

Dehydrated Luria Bertani (LB) broth (DF0446-17-3) was purchased from Thermo Fisher (Difco^TM^, Pittsburgh, PA, USA). Commercially available cyanoacrylate monomers, e.g., Aron Alpha 201, 501 (Toagosei, Aichi, Japan), and TB7721 (Three Bond, Ehime, Japan) were used as materials for ethyl cyanoacrylate, isobutyl cyanoacrylate, and ethoxyethyl cyanoacrylate, respectively. Tween 80 (160-21772) was obtained from Fujifilm (Osaka, Japan). All other reagents used were of analytical grade.

### 2.2. Bacterial Strains and Culture Conditions

Five Gram-positive bacterial strains, *Bacillus subtilis* (*B. subtilis* NBRC 13719), *Brevibacillus agri* (*B. agri* NBRC 15538), *Microbacterium aurum* (*M. aurum* NBRC 15204), *Propionibacterium acnes* (*P. acnes* NBRC 107605), and *Staphylococcus aureus* (*S. aureus* NBRC 12732), and five Gram-negative bacteria, *Escherichia coli* (*E. coli* NBRC 3301), *Klebsiella pneumonia* (*K. pneumoniae* NBRC 13277), *Pseudomonas aeruginosa* (*P. aeruginosa* NBRC 3080), *Salmonella typhimurium* (*S. typhimurium* NBRC 13245), and *Serratia marcescens* (*S. marcescens* NBRC 102204) were obtained from National Institute of Technology and Evaluation (NITE) Biological Resource Center (Tokyo, Japan). All the strains were cultured in Luria Bertani nutrient solution (LB) at 30 °C in a rotary shaker at 95 rpm agitation, except for *S. marcescens*, which was incubated at 25 °C.

### 2.3. Synthesis of PACA Nanoparticles

We synthesized three different types of PACA-NPs, ethoxyethyl cyanoacrylate-NPs (EECA-NPs), ethyl cyanoacrylate-NPs (ECA-NPs), and isobutyl cyanoacrylate-NPs (*^i^*BCA-NPs), using the corresponding monomers described above. In brief, PACA-NPs were synthesized using a modified method reported by Shirotake et al. [25]. Monomer solutions were added dropwise into 5 N HCl containing 4% (*v/v*) Tween 80 until the added monomer reached 4% (*v/v*) of the HCl solution under continuous stirring at 250 rpm at 25 °C. After stirring for 2 h to aggregate polymer particles, 0.5 N NaOH was added drop wisely until the pH reached 7.0, and stirred for an additional hour to generate solid nanoparticles. The solution was filtered to remove any debris in the solution, if present, using a 5 μm pore size membrane filter (SMWP04700, Millipore, Merck, Darmstadt, Germany). The synthesized PACA-NPs were stored at 4 °C until further use.

Zeta sizer ELSZ-2000ZS instrument (Otsuka Electronics, Osaka, Japan) was used to determine the NPs’ size distribution based on the intensity of scattered light and zeta potentials. Each value represents the mean of at least three independent experiments.

### 2.4. Growth Inhibition Zone Assay

To evaluate the cell growth inhibitory effect, the bacterial cells were spread on agar plates to obtain a confluent monolayer. Paper disks soaked with various concentrations of PACA-NPs were added to each agar plate and tested using Kirby–Bauer disk diffusion method [26,27,28]. The size of the zone without bacterial growth (halo) was measured: Briefly, 5 μL of PACA-NPs suspensions (final concentration, 10–1000 mg/L) were directly spotted on 6 mm sterile paper disks (No. 3 Whatman filter paper, Millipore) and placed on the surface of LB agar plates containing bacterial culture (~10^6^ CFU/mL). The halo diameters were measured after overnight incubation at 30 °C. An amount of 0.01% (*v/v*) Tween 80 was employed as a negative control, which is the dispersant contained in the synthesized PACA-NPs’ suspensions. The measurements were taken from three independent experiments.

### 2.5. Analysis Cell Growth Inhibition in Liquid Medium by Measuring Turbidity

To confirm the growth inhibitory effect of PACA-NPs, optical density (OD) changes in the LB culture containing PACA-NPs at 100 mg/L and Tween 80 only (control) were measured, to reflect cell proliferation changes. Briefly, overnight bacterial culture of approximately 1 × 10^9^ CFU/mL was diluted 10^4^-fold into 5 mL of LB in a test tube containing PACA-NPs at 100 mg/L or only 0.01% (*v/v*) Tween80. These tubes were incubated at 30 °C with 95 rpm shaking. The OD was recorded every hour for 10 h using McFarland Densitometer (DEN-1B, Grant instruments, Riga, Latvia). All experiments and their respective control samples were performed in triplicates for each strain.

### 2.6. Effect of N-acetyl-L-Cysteine on Cell Growth Inhibition by ECA-NPs Exposure

Considering that generation of ROS is a possible cause for cell growth inhibitions by PACA-NPs, we used N-acetyl-L-cysteine (NAC; 017-0513, Fujifilm, Osaka, Japan), a well-known ROS scavenger, as antioxidant and precultured cells with it. Bacterial cells were cultured for one hour in the LB medium containing 0, 2, 4, or 8 mM of NAC prior to the addition of 100 mg/L ECA-NPs. Moreover, LB medium containing only 4 mM NAC was used to detect the toxicity of NAC in the absence of ECA-NPs. Immediately after addition of ECA-NPs (time 0), the OD was monitored hourly for 10 h. Each experiment was performed independently in triplicate.

### 2.7. Fluorescence Microscopic Observation of Membrane Damage

To analyze cell membrane disturbance after ECA-NPs treatment, the untreated and treated bacterial cells (1 × 10^5^ cells/mL of LB) were harvested. Briefly, after incubation with ECA-NPs at 100 mg/L for 2 h at 30 °C, 5 μL of propidium iodide (PI) was added to each sample and then incubated for 1 h in the dark at room temperature. An amount of 5 μL of the treated samples and control were observed using Olympus BX53 microscope (Evident, Tokyo, Japan) with the 100× objective lens and a fluorescence filter set. Images were acquired and analyzed using Olympus cellSens software (cellSens V3.2, Tokyo, Japan).

## 3. Results

### 3.1. Characteristics of the Synthesized PACA-MPs

The average sizes of the PACA-NPs were 82.5 ± 0.42 nm, 248 ± 2.0 nm, and 22.9 ± 0.46 nm for ECA-NPs, EECA-NPs, and *^i^*BCA-NPs, respectively. All PACA-NPs used in this study were synthesized using a neutral detergent, Tween 80, as a dispersant. Under these conditions, EECA-NPs had a much larger negative zeta potential (−65.9 ± 3.53 mV) compared to ECA-NPs (−6.72 ± 2.78 mV) and *^i^*BCA-NPs (−1.91 ± 4.11 mV) (Table 1).

### 3.2. Cell Growth Inhibition Activity of PACA-NPs

To determine the growth inhibitory effect of the synthesized PACA-NPs, we performed both solid- and liquid-based growth studies in the selected bacteria.

#### 3.2.1. Growth Inhibition Zone Observation

Growth inhibition zones that reflect the susceptibility of bacteria to the NPs were measured after overnight incubation at 30 °C according to the Kirby–Bauer test protocol [28]. There were distinct circular clear zones (halos) of various sizes around the paper disks depending on the NPs used, which were not detected in the control disks with Tween 80. The corresponding diameters of growth inhibition zones are summarized in Table 2. The halos produced by PACA-NPs against the selected Gram-positive and Gram-negative bacteria are also shown in Appendix A. The halo diameters increased with an increase in the concentration of the absorbed PACA-NPs. ECA-NPs showed significant antibacterial activity against all tested bacteria species, with larger halos observed against Gram-positive than the Gram-negative bacteria, except Gram-positive *P. acne*, where smaller or no halos were observed compared to the tested Gram-negatives, even at the higher concentrations (Appendix A).

The effect of EECA-NPs was rather moderate compared to ECA-NPs against both Gram-positive and Gram-negative bacteria. Moreover, its effect against the Gram-positive species was similar to that of *^i^*BCA-NPs, without any effect against the Gram-positive bacterium *P. acne*. However, EECA-NPs have slight but detectable inhibitory effects against Gram-negative bacteria at 100 or 1000 mg/L (Table 2). Conversely, *^i^*BCA-NPs showed antibacterial activity only against Gram-positive bacteria and no inhibitory effect against all Gram-negative bacteria even at 1000 mg/L (Table 2).

#### 3.2.2. Effect of ECA-NPs on the Bacterial Growth Dynamics

Furthermore, we investigated the cell growth inhibition effect in LB medium containing PACA-NPs at 100 mg/L. Bacterial growth was monitored by measuring the turbidity change over time. In the control culture containing only 0.01% (*v/v*) Tween 80, all bacterial species displayed similar growth profiles to those seen in LB medium only, indicating the negligible toxicity of the Tween 80.

The growth of three Gram-positive bacteria, (i.e., *B. subtilis*, *M. aurum*, and *S. aureus*; Figure 1A,C,E, respectively) and two Gram-negative bacteria, (i.e., *E. coli* and *S. typhimurium*; Figure 1F,I, respectively), was completely inhibited over a 10 h period in the LB medium containing 100 mg/L ECA-NPs. Conversely, growth inhibition was limited to 4–6 h in two Gram-positive bacteria, (i.e., *B. agri*, *P. acnes*; Figure 1B,D, respectively), and in three Gram-negative bacteria, (i.e., *K. pneumonia*, *P. aeruginosa*, *S. marcescens*; Figure 1G,H,J, respectively).

EECA-NPs exposure at the same concentrations also induced substantial growth inhibition for 10 h in only two Gram-positive species, *B. subtilis* and *S. aureus* (Figure 1A,E, respectively). Clearly delayed cell growth was observed in all examined Gram-negative bacteria, *E. coli*, *K. pneumonia*, *P. aeruginosa*, *S. typhimurium*, and *S. marcescens* (Figure 1F–J, respectively), while no significant cell growth inhibition was observed in three Gram-positive bacteria, *B. agri*, *M. aurum*, *P. acnes* (Figure 1B–D, respectively).

After *^i^*BCA-NP exposure, the growth of two Gram-positive bacteria, i.e., *B. subtilis* and *S. aureus* (Figure 1A,E, respectively), was restored after 5–7 h of exposure, while it had almost no inhibitory effect on all five Gram-negative bacteria as their growth rates were very similar to those of the untreated control.

Thus, ECA-NPs were more effective than EECA-NPs and *^i^*BCA-NPs against all ten bacterial species. EECA-NPs had slightly stronger effect than *^i^*BCA-NPs against six bacterial species. Notably, EECA-NPs were more effective against Gram-negative bacteria than Gram-positive bacteria. Moreover, the efficacy of EECA-NPs and *^i^*BCA-NPs were equally poor against for three Gram-positive bacterial species, i.e., *B. agri*, *M. aurum*, *P. acnes*, and one Gram-negative bacterial species *K. pneumonia*.

### 3.3. Effect of ROS Scavenger NAC on Cells after Preliminary Treatment

Based on a previous study showing that exposure to 100 mg/L of *^i^*BCA-NPs induced ROS accumulation in a green alga *C. reinhardtii* [22], we investigated whether cells pretreated with NAC, by culturing in a medium containing NAC, alleviates the inhibition of cell growth caused by ECA-NPs exposure. Of the three synthesized PACA-NPs, ECA-NPs treatment was selected as it exhibited the highest cell growth inhibitory effect on both Gram-positive and Gram-negative bacteria.

As shown in Figure 2, in all bacterial species, co-incubation with NAC at 2–8 mM constantly showed higher growth rates than without NAC. However, the optimal NAC concentration varies from species to species, i.e., 4 mM NAC was sufficient for *B. subtilis*, *P. acnes*, and *S. aureus*, while for *B. agri* and *K. pneumonia*, growth could only be restored with 8 mM of NAC. Moreover, when *E. coli* and *S. typhimurium* were co-cultured with 2 or 4 mM of NAC, their growth rates were not significantly different from the control without ECA-NPs. Notably, the growth of *B. subtilis*, *B. agri, S. aureus*, and *S. marcescens*, in LB containing 4 mM NAC was slower than that in LB only, suggesting possible toxicity of NAC at concentrations exceeding 4 mM in these species.

### 3.4. Fluorescent Microscopy Examination of Cytoplasmic Membranes

We verified whether ECA-NPs exposure caused any damage to the cell membrane in *B. subtilis* and *E. coli*, representing Gram-positive and Gram-negative bacteria, respectively, using red fluorescent probe PI that only binds to DNA and RNA in cells with compromised cell membranes, i.e., dead cells. As shown in Appendix A, strong red fluorescence was detected in both NP-treated *B. subtilis* and *E. coli*, indicating that exposure to ECA-NPs at 100 mg/L for 2 h caused severe damage to the cytoplasmic membrane leading to cell death. This was supported by the lack of cell growth observed in these species after co-incubation with 100 mg/L ECA-NPs for 10 h (Figure 2).

## 4. Discussion

To our knowledge, the information available on the growth inhibitory mechanism induced by PACA-NPs and the spectrum of bacterial species sensitive to various PACA-NPs is limited [29,30]. To explore the use of PACA-NPs as potential antibacterial agents, we studied its growth inhibitory effects against five Gram-positive and five Gram-negative bacterial species using a growth inhibition zone assay and concurrently monitoring the growth dynamics in the LB medium. We used the growth inhibition zone method to obtain preliminary data as it is simpler and can be used for testing a large number of samples before conducting a cell growth inhibition assay in a liquid medium.

This assay showed that, of the three PACA-NPs, ECA-NPs were most effective against all the bacterial strains, showing a higher inhibitory effect against Gram-positive bacteria than Gram-negative bacteria. Similar results were shown by Manzano et al. (2006), regardless of the synthesis or application method of the polymers to the bacterial cells [31]. In their study, one drop of ethyl cyanoacrylate monomer was added directly onto an agar plate to investigate its antibacterial effects against six types of bacteria. Among tested bacterial strains, the ethyl-cyanoacrylate exhibited inhibitory effects against four Gram-positive bacterial species (*Enterococcus faecalis*, *S. aureus, Streptococcus pyogenes*, and *Streptococcus pneumoniae*) and one Gram-negative bacterium (*E. coli*). However, no inhibition zones were observed for the Gram-negative bacterium *P. aeruginosa*.

Interestingly, our results showed that paper disks containing a 5 μL solution of 100 or 1000 mg/L synthesized ECA-NPs inhibited the growth of *P. aeruginosa* (Table 2, Appendix A), while Manzano et al. (2006) reported no antibacterial effect of ECA-polymers against the same strain [31]. This discrepancy might be due to the different methodologies used: Manzano et al. (2006) loaded ECA monomer solution directly on the agar where it was polymerized [31], while in our study, paper disks soaked with ECA-NPs solution were placed onto the agar plates.

Another study conducted by Rushbrook et al. (2014) showed that when solidified pellets of the octyl cyanoacrylate were directly loaded on agar plates, they inhibited the growth of the three Gram-positive bacteria, *S. aureus*, *Staphylococcus* sp., and *Streptococcus* sp., but did not inhibit growth in the two Gram-negative bacteria, *E. coli* and *P. aeruginosa* [32]. Considering our results and theirs, the spectrum of bacteria sensitive to octyl cyanoacrylate pellets is rather similar to that of *^i^*BCA-NPs.

Here, the growth inhibition assay results using liquid media were consistent with the growth inhibition zone assay on agar plates, except in the case of Gram-positive bacteria *M. aurum*. In the growth inhibition zone assay, paper disks containing either ECA-, EECA-, or *^i^*BCA-NPs clearly showed halos around the disks (Table 2, Appendix A). However, when growth inhibition was studied in LB liquid medium, the presence of EECA- or *^i^*BCA-NPs at a concentration of 100 mg/L appeared to be less effective than on agar plates. The reason for this discrepancy is unclear. To date, there are no studies showing differences in the mechanisms of cell growth inhibition of nanoparticles in liquid media and on the surface of agar plates. We hypothesize that in the liquid phase, nanoparticles stripped the cell surface proteins upon their collision, which might initiate cell death [22,23].

Romero et al. (2009) compared the effect of direct synthesis of ECA-polymer and N-butyl-cyanoacrylate (*^n^*BCA)-polymer on agar plate by both adding these monomers directly onto the agar plate and using paper disks soaked with them. They observed that ECA-polymers were more effective against *E. coli* than the *^n^*BCA-polymer [29], and suggested that this might be due to the different degradation rates of these polymers accompanied by the release of formaldehyde and cyanoacetate [33]. ECA-polymers are expected to degrade faster than *^n^*BCA-polymers due to the smaller alkyl side groups in ECA-polymers, (i.e., ethyl-group) than *^n^*BCA-polymers, (i.e., butyl-group).

Based on both the growth inhibition investigations we conducted, the NPs in order of increasing efficacy are ECA-, EECA-, and *^i^*BCA-NPs for all species. However, this order is not on the basis of the size of the alkyl side group, i.e., ethyl, ethoxyethyl, and isobutyl for ECA-NPs, EECA-NPs, and *^i^*BCA-NPs, respectively. The smaller the alkyl side group, the faster the degradation of the NPs and hence, the higher the amount of formaldehyde released. Hence, the predicted order of effectiveness will be ECA-, *^i^*BCA-, and EECA-NPs. This suggests that the degradation rate of these nanoparticles does not depend solely on the size of the side chain alkyl groups [34] or that the cell growth inhibition rate is not dependent on the amount of released formaldehyde.

ROS is a by-product of respiration in cells and the excessive production of ROS has been shown to trigger or directly cause cell death [35,36,37]. We suggested that the ROS generation might be the main cause of cell growth inhibition because pre-treatment of cells with appropriate concentrations of NAC significantly recovered the cell growth rate to that of the control (Figure 2). This was in accordance with the results of Song et al. (2019) showing that ROS induced cell death due to exposure to silver/curcumin composite NPs in *B. subtilis* and *E. coli* was reduced after adding NAC [38]. Therefore, it is important to understand how these particles promote ROS generation to elucidate the mechanism of growth inhibition by PACA-NPs.

Among the three kinds of nanoparticles used in this study, the mean average diameter of the *^i^*BCA-NPs was 22.9 nm, which is prominently smaller than ECA-NPs and EECA-NPs are 82.5 nm and 248 nm, respectively. Small-sized metal nanoparticles have been shown to be more effective in inducing cell death or growth inhibition as they can penetrate or diffuse into the cells better [39,40,41]. Interestingly, our results showed that the inhibitory effect of PACA-NPs was not dependent on the size of the particles, as the smallest NPs, *^i^*BCA-NPs, were, in fact, the least efficient (Table 1, Figure 1).

Analysis of particle size distribution of the *^i^*BCA-NPs showed that the size of the smallest particle contained in the *^i^*BCA-NPs was approximately 10 nm (Appendix A). However, the reported pore sizes of the bacterial cell wall are 2.06 nm in *E. coli* and 2.12 nm in *B. subtilis* [24]. This suggests that all the nanoparticles used here cannot easily pass through without significantly damaging the cell wall. Therefore, even the smallest particles contained in the *^i^*BCA-NPs cannot pass through the outer membrane via specific channels or classical porins embedded in it, as their pore sizes are ca. 0.5–0.6 nm in diameter [42,43]. It is likely that NPs need to disrupt the integrity of the outer membrane of Gram-negative bacteria, before interacting with the peptidoglycan layer if it is the cell growth inhibition target.

Considering the above, the molecules that NPs can possibly interact with are molecules embedded in the outermost structure of the cell wall. Shirotake et al. (2014) hypothesized that *^n^*BCA-NPs can induce cell lysis of Gram-positive bacteria by direct adhering to the peptidoglycan layer of the cell wall and that this attachment induces an uneven cell wall synthesis [30]. Conversely, Widyaningrum et al. (2019) showed that exposure to *^i^*BCA-NPs at 100 mg/L acutely induced cell death in a unicellular green alga *Chlamydomonas reinhardtii* [22]. The study claimed that the collision of the NPs with the cell surface might inactivate or remove proteins located on the cell wall to induce cell death. This might be accompanied by ROS generation and hydrolysis of hydroxy-proline-rich glycoproteins in the cell wall [22].

The cell wall structure is known to differ between Gram-positive and Gram-negative bacteria. In Gram-negative bacteria, its outermost layer is an additional lipid bilayer mainly composed of (lipo)proteins, phospholipids, and anchored-lipopolysaccharides [44]. In Gram-positive bacteria, instead of this outer membrane, only a thick peptidoglycan layer exists throughout the cell walls [45]. Despite this difference in the outermost structure, ECA-NPs inhibited cell growth in both bacterial types, while EECA-NPs tended to be more effective against Gram-positive bacteria than Gram-negative bacteria. Whereas *^i^*BCA-NPs were effective against only some Gram-positive bacteria. Thus, the efficacy spectrum of these nanoparticles in bacteria is complex and it is unlikely that their effect is determined by differences in the cell wall structure in Gram-positive and Gram-negative bacteria.

Some bacterial species have been shown to form capsules, collagenous membrane-like structures composed of polysaccharides or polypeptides, surrounding the cell wall [46]. The presence or absence of a capsule does not correspond to the classification of Gram-positive or Gram-negative bacteria. Moreover, even within the same species, these capsules might be absent due to slight differences in culture conditions [46]. Hence, the complex spectrum of cell growth inhibitory effects by *^i^*BCA-NPs might be attributed to the attachment of these NPs to the capsule.

We used PI staining to indicate cell death as PI can only penetrate cells with disrupted membrane integrity, which leads to cell death [37]. In *B. subtilis* and *E. coli*, cells exposed to 100 mg/L ECA-NPs were found to take up the PI and emit red fluorescence (Appendix A). Moreover, these cells did not proliferate at all for 10 h over the entire period of measurement, indicating that treatment with 100 mg/L ECA-NPs not only inhibits cell growth but also induces cell death in these bacteria. Moreover, it is likely that cell death is also induced in *M. aurum* and *S. aureus*, where no cell proliferation was observed in the liquid culture for 10 h after the same treatment. Cell death due to ROS accumulation associated with cell membrane damage is known to occur in various bacteria and yeasts [47,48].

The zeta potentials of EECA-NPs, ECA-NPs, and *^i^*BCA-NPs were all negative, approximately −66 mV, −6.7 mV, and −1.9 mV, respectively (Table 1). The largest negative charge potential of EECA-NPs must be due to the chemical characteristics of the side group. It is possible that the difference in the extent of negative zeta potential affects how strongly cell growth is inhibited. Zeta potential can be controlled by changing the surfactants used in the synthesis reaction or salts in the culture medium. Therefore, NPs synthesized from the same monomer might have different charges. Further investigations are required to show the effect of NPs’ zeta potential on the cell growth inhibitory ability.

## 5. Conclusions

The current study shows a potent antibacterial activity of nanoparticles of PACA against both Gram-positive and Gram-negative bacteria. In general, ECA-NPs were significantly toxic against Gram-positive and Gram-negative bacteria strains compared to EECA-NPs and *^i^*BCA-NPs. However, the inhibitory profiles of these NPs on bacterial cell growth are complex. Regardless of the type of bacteria or NPs used, the addition of NAC restored the cell growth rates to similar levels to that of the culture without NPs. In addition to the above results, the antibacterial activity of PACA-NPs against Gram-positive and Gram-negative bacteria may also involve disrupting the membrane integrity through the generation of ROS. Therefore, the generated ROS is likely to be the main cause of the cell growth inhibition induced by these PACA-NPs.

## Figures and Tables

**Figure 1 membranes-12-00782-f001:**
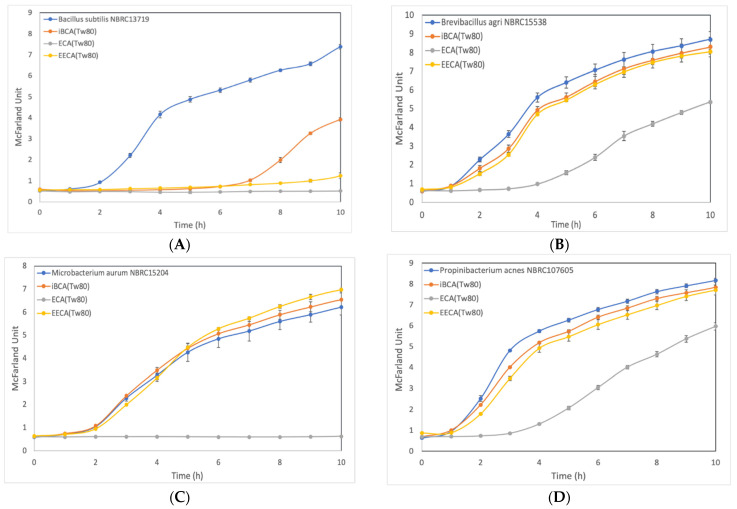
Bacterial growth cultures exposed to PACA-NPs. Control (blue line), growth in LB medium containing 0.01% Tween 80. All cultures were set up and grown under the same conditions as described in Materials and Methods. The growth was shown by McFarland unit. (**A**), *Bacillus subtilis* (Gram-positive); (**B**), *Brevibacillus agri* (Gram-positive); (**C**), *Microbacterium aurum* (Gram-positive); (**D**), *Propionibacterium acnes* (Gram-positive); (**E**), *Staphylococcus aureus* (Gram-positive); (**F**), *Escherichia coli* (Gram-negative); (**G**), *Klebsiella pneumonia* (Gram-negative); (**H**), *Pseudomonas aeruginosa* (Gram-negative); (**I**), *Salmonella typhimurium* (Gram-negative); (**J**), *Serratia marcescens* (Gram-negative). The error bars represent standard deviations determined from at least three duplicates.

**Figure 2 membranes-12-00782-f002:**
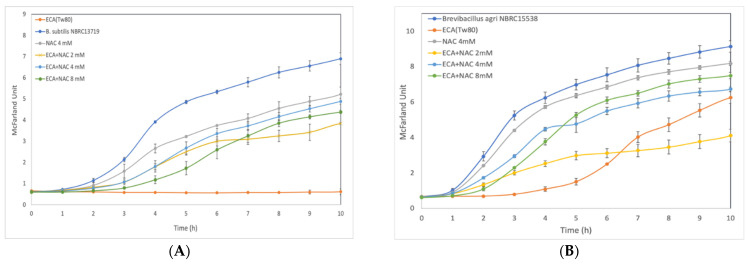
Effect of co-incubation with N-acetyl-L-cysteine for the cell growth inhibition. ECA-NPs exposure at 100 mg/L was carried out in the LB containing various concentration of NAC. The growth was shown by McFarland unit. (**A**), *Bacillus subtilis* (Gram-positive); (**B**), *Brevibacillus agri* (Gram-positive); (**C**), *Microbacterium aurum* (Gram-positive); (**D**), *Propionibacterium acnes* (Gram-positive); (**E**), *Staphylococcus aureus* (Gram-positive); (**F**), *Escherichia coli* (Gram-negative); (**G**), *Klebsiella pneumonia* (Gram-negative); (**H**), *Pseudomonas aeruginosa* (Gram-negative); (**I**), *Salmonella typhimurium* (Gram-negative); (**J**), *Serratia marcescens* (Gram-negative). The data represent the average and standard deviation from three independent experiments.

**Table 1 membranes-12-00782-t001:** Characteristics of synthesized resin cyanoacrylate nanoparticles.

	Size (nm)	Zeta Potential (mV)
EECA-NPs	248 ± 2	−65.9 ± 3.53
ECA-NPs	82.5 ± 0.42	−6.72 ± 2.78
*^i^*BCA-NPs	22.9 ± 0.46	−1.91 ± 4.11

The data are expressed as means of ± SD of three independent measurements.

**Table 2 membranes-12-00782-t002:** Growth inhibition zone exerted by PACA-NPs dipped paper discs.

Micro Organism	Species	PACA-NPs									C ^a^
		ECA-NPs (mg/L)			EECA-NPs (mg/L)			*^i^*BCA-NPs (mg/L)			
		10	100	1000	10	100	1000	10	100	1000	
Gram-positive bacteria	*Bacillus subtilis*	7	9	11.5	n.a.	6.5	12.5	7	7.5	10	n.a.
	*Brevibacillus agri*	7.5	8.5	11	n.a.	n.a.	7.5	n.a.	n.a.	n.a.	n.a.
	*Microbacterium aurum*	9	10	16	7	9	15	8	10	12	n.a.
	*Propionibacterium acnes*	n.a.	n.a.	7.5	n.a.	n.a.	n.a.	n.a.	n.a.	6.5	n.a.
	*Staphylococcus aureus*	n.a.	10	14	n.a.	7	10	n.a.	7	10	n.a.
Gram-negative bacteria	*Escherichia coli*	6.5	9	10	n.a.	6.5	7.5	n.a.	n.a.	n.a.	n.a.
	*Klebsiella pneumoniae*	n.a.	9	12	n.a.	n.a.	7	n.a.	n.a.	n.a.	n.a.
	*Pseudomonas aeruginosa*	n.a.	8	11	n.a.	n.a.	6.5	n.a.	n.a.	n.a.	n.a.
	*Serratia marcescens*	7	8.5	11	n.a.	n.a.	6.5	n.a.	n.a.	n.a.	n.a.
	*Salmonella typhimurium*	7	10	10	n.a.	8.5	9	n.a.	n.a.	n.a.	n.a.

^a^: Result of the control experiments in which 6 mm paper discs congaing 0.01% (*v/v*) Tween 80 were used. n.a.: not active.

## Data Availability

The data that supports the findings of this study are available in the Appendix A of this article.

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
