# Peer review of "Evaluation of the Growth-Inhibitory Spectrum of Three Types of Cyanoacrylate Nanoparticles on Gram-Positive and Gram-Negative Bacteria"

_membranes, 2022, doi:10.3390/membranes12080782_

Round 1

Reviewer 1 Report

The manuscript entitled “Evaluation of the growth-inhibitory spectrum of three types of cyanoacrylate nanoparticles on Gram-positive and -negative bacteria” submitted to membrane is a reasonable document and deals with NPs as antibacterial agents. Generally, presentation of the work being carried out is fairly acceptable, it needs several changes of major nature. Some specific comments are given hereunder;

1.     There are several linguistic issues which need to be fixed such as,

“……other reagents used were of analytical grade used.”

“….similar growth profile as that seen in……”

“This assay showed that……… in this study”

“this suggests that this…..”

 2.     Check the entire text for uniformity such as space between the respective value and °C. There must be no space as per standard rules. Other units may also be verified in light of international standard. 

3.     What type of available standard was used during the study? If not, then how authors would validate their results?

4.     Species name must be full at their first mention followed by abbreviated (first name) names. Like E. coli etc.

5.     There are some other in-field strategies to improve efficiency of either the existing antibacterial medicines or to introduce novel natural products against MDR bacterial strains, they are worth mentioning in discussion. Some of the articles are suggested  for reading here; 10.3390/antibiotics11060797; 10.3390/antibiotics10080939; 10.1038/s41598-021-92622-0; 10.1007/s10876-019-01688-4 and DOI: 10.1007/s11051-020-04939-y.

6.     Figure 3 is missing in the text.

7.     For normal- and iso-butyl derivatives, the n and i are better to be superscripts and italic. Like iB and  nB 

8.     Conclusion of MS may be polished and quantitative data must be included.

Author Response

The manuscript entitled “Evaluation of the growth-inhibitory spectrum of three types of cyanoacrylate nanoparticles on Gram-positive and -negative bacteria” submitted to membrane is a reasonable document and deals with NPs as antibacterial agents. Generally, presentation of the work being carried out is fairly acceptable, it needs several changes of major nature. Some specific comments are given hereunder:

1. There are several linguistic issues which need to be fixed such as,

“……other reagents used were of analytical grade used.”

“….similar growth profile as that seen in……”

“This assay showed that……… in this study”

“this suggests that this…..”

Response: 

Manuscript has been revised accordingly.

“……other reagents used were of analytical grade.”

“……similar growth profiles to those seen in ……”

“This assay showed that, of the three PACA-NPs, ……”

“……, and suggesting that this ……”

2. Check the entire text for uniformity such as space between the respective value and °C. There must be no space as per standard rules. Other units may also be verified in light of international standard.

Response: Manuscript has been revised accordingly. Thank you.

3. What type of available standard was used during the study? If not, then how authors would validate their results?

Response: 

In this study, we have also examined toxic effect of Tween 80, as dispersant. The growth curve of all strains under normal condition (NPs free) and Tween 80 clearly depicted the lag, log and stationary phase. The results then indicated that addition of PACA-NPs on the cultures are linked to the observed cytotoxicity.

Besides that, each experiment was independently performed in triplicate.

4. Species name must be full at their first mention followed by abbreviated (first name) names. Like E. coli etc.

Response:

Manuscript has been revised accordingly. Thank you.

5. There are some other in-field strategies to improve efficiency of either the existing antibacterial medicines or to introduce novel natural products against MDR bacterial strains, they are worth mentioning in discussion. Some of the articles are suggested for reading here; 10.3390/antibiotics11060797; 10.3390/antibiotics10080939; 10.1038/s41598-021-92622-0; 10.1007/s10876-019-01688-4 and DOI: 10.1007/s11051-020-04939-y.

Response:

Thank you for the suggestion.

We included the sentence below in response to the comment in the revised manuscript.

“The antibacterial activity of NPs depends on various factors such as their chemical composition, shape and size [8]. With significant developments in the understanding of nanosystems, some strategies to enhance NPs’ activity has been assessed, including combining treatment of NPs and irradiation [9], coating NPs with several coating agents [10], and conjugating NPs with antibiotics [11,12].”

Refs:

8. Sukhanova, et al., 2018, Dependence of nanoparticle toxicity on their physical and chemical properties, Nanoscale Res Lett, 13.

9. Al-Sharqi, et al., 2019, Enhancement of the antibacterial efficiency of silver nanoparticles against gram-positive and gram-negative bacteria using blue laser light, Int J Photoenergy, 2019.

10. Jaworski, S., 2018, Graphene oxide-based nanocomposites decorated with silver nanoparticles as an antibacterial agent, Nanoscale Res Lett, 13.

11. Ali, S., et al., 2020, Nano-conjugates of Cefadroxil as Efficient Antibacterial Agent Against Staphylococcus aureus ATCC 11632, 31: 811-821.

12. Ali, S., et al., 2020, Bactericidal potentials of silver and gold nanoparticles stabilized with cefixime: a strategy against antibiotic-resistant bacteria, J. Nanoparticle Res, 22.

6. Figure 3 is missing in the text.

Response: Figure 3 has been added to the revised manuscript.

7. For normal- and iso-butyl derivatives, the n and i are better to be superscripts and italic. Like iB and nB.

Response: 

Manuscript has been revised accordingly. Thank you.

8. Conclusion of MS may be polished and quantitative data must be included.

Response: 

We included the sentence below in response to the comment in the revised manuscript. The current study shows a potent antibacterial activity of nanoparticles of PACA against both Gram-positive and -negative bacteria. In general, ECA-NPs were significantly toxic against Gram-positive and -negative bacteria strains compared to EECA-NPs and iBCA-NPs. However, the inhibitory profiles of these NPs on bacterial cell growth are complex. Regardless of the type of bacteria or NPs were used, the addition of NAC restored the cell growth rates to similar levels to that of the culture without NPs. In addition to the above results, the antibacterial activity of PACA-NPs against Gram-positive and -negative bacteria may also involve disrupting the membrane integrity through the generation of ROS. Therefore, the generated ROS is likely to be the main cause for the cell growth inhibition induced by these PACA-NPs.

Reviewer 2 Report

The manuscript presented for review contains interesting results. However, before it can be considered for publication, the authors need to revise their work, my main observations being:

1. The polymer nanoparticles needs a more thorough characterisation. Molecular weight should be determined. Also, SEM/TEM analysis should be performed.

2. The dimensions recorded for EECA (248 nm) does not qualify the material as nanoparticles.

3. Row 133 - the negative sign is not presented for the zeta potential of EECA.

4. Table 2 - the values of 6 mm should be replaced by not active, in order to better understand the results

5. Some minor typos should be corrected. Please use the degree sign for the temperature values. 

Author Response

The manuscript presented for review contains interesting results. However, before it can be considered for publication, the authors need to revise their work, my main observations being:

1. The polymer nanoparticles needs a more thorough characterisation. Molecular weight should be determined. Also, SEM/TEM analysis should be performed.

Response: 

We would like to thank you for the suggestions.

The molecular weight data and SEM analysis of PACA-NPs have been described in our previous reports (see Refs). Since, so far, we know, the relationship between particle size and the antibacterial activity has been widely reported and the particle size has been recognized as a key factor in the application of NPs. In addition, we are unable to carry out other MW determination and SEM analysis by ourselves due to the absence of the necessary equipment as well as the limited time given for this revision.

Therefore, we can only include size distribution and zeta potential in this study.

Refs:

  • Miyashita, R., et al., 2019, Synthesis and characterization of cyanoacrylate nanoparticles in aqueous media, Proceed 68th Soc Sci Jap Ann Meeting.
  • Widyaningrum, D., et al., 2018, Acutely induced cell mortality in the cellular green alga Chlamydomonas reinhardtii (Chlorophyceae) following exposure to acrylic resin nanoparticles, J Phycol, 55 (1): 118-133.

2. The dimensions recorded for EECA (248 nm) does not qualify the material as nanoparticles.

Response:

Thank you for this comment.

We understand that a nanoparticle is usually defined as a particle with a size in the range of 1 to 100 nanometers in diameter. However, the term itself sometimes can be used for larger particles (Wyatt, 2018), as nanoparticles can clump together that will have outside dimensions larger than 100 nm.

Some studies also have considered nanoparticles with dimensions > 100 nm, for example:

Chenthamara, et al. (2019) reviewed various nanocarriers (nanoparticles) to improve drug delivery. In this review, the synthetic polymeric nanoparticles that are often studied and used, typically have size between 20 and 250 nm.

While for some purposes, the particle diameter ideally should be 10-150 nm.

Li, et al. (2017) investigated Fe3O4 NPs with sizes 10-300 nm to reveal the fundamental relationship between the crystal domain structure and the magnetic properties.

Liyuan, et al. (2013) investigated the potency of size-controlled mesoporous titanium dioxide (TiO2) nanoparticles, with size: particle A, 301 ± 160 nm; particle B, 264 ± 99 nm.

Based on these publications, the authors considered using EECA-NPs is still valuable.

Refs:

  • Chenthamara, D., et al., 2019, Therapeutic efficacy of nanoparticles and routes of administration, Biomaterials Res, 23 (20).
  • Wyatt, P. J., 2018, Measuring nanoparticles in the size range to 2000 nm, J Nanoparticle Res, 12.
  • Li, Q., et al., 2017, Correlation between particle size/domain structure and magnetic properties of highly crystalline Fe3O4 nanoparticles, Sci Rep, 7: 9894.
  • Liyuan, H., et al., 2013, Mesoporous TiO2 nanoparticles: A new material for biolistic bombardment, Phycological Res, 61 (1): 58-60.

3. Row 133 - the negative sign is not presented for the zeta potential of EECA.

Response: We have corrected it; negative sign has been added for the zeta potential of EECA.

4. Table 2 - the values of 6 mm should be replaced by not active, in order to better understand the results.

Response: Considering the comment, manuscript has been revised accordingly.

5. Some minor typos should be corrected. Please use the degree sign for the temperature values.

Response: Manuscript has been revised accordingly. Thank you.

Round 2

Reviewer 1 Report

I don't recommend Fig. 3 to me incorporated in the main manuscript, further this reviewer is satisfied from most of the changes being made by the authors and has no objection regarding acceptance of the article in its present form. 

Reviewer 2 Report

The authors improved the manuscript according the reviewers' suggestion.